# *m*-Carborane as a Novel Core for Periphery-Decorated Macromolecules

**DOI:** 10.3390/molecules25122814

**Published:** 2020-06-18

**Authors:** Ines Bennour, Francesc Teixidor, Zsolt Kelemen, Clara Viñas

**Affiliations:** Institut de Ciència de Materials de Barcelona (ICMAB-CSIC), Campus UAB, 08193 Barcelona, Spain; bennourines@ymail.com (I.B.); teixidor@icmab.es (F.T.); kelemen.zsolt@mail.bme.hu (Z.K.)

**Keywords:** *m*-carborane, electrophilic substitution, coupling reaction, organic branches, Hirshfield Study

## Abstract

*Closo**m*-C_2_B_10_H_12_ can perform as a novel core of globular periphery-decorated macromolecules. To do this, a new class of di and tetrabranched *m*-carborane derivatives has been synthesized by a judicious choice of the synthetic procedure, starting with 9,10-I_2_-1,7-*closo*-C_2_B_10_H_10_. The 2a-NPA (sum of the natural charges of the two bonded atoms) value for a bond, which is defined as the sum of the NPA charges of the two bonded atoms, matches the order of electrophilic reaction at the different cluster bonds of the icosahedral *o*-and *m*- carboranes that lead to the formation of B-I bonds. As for *m*-carborane, most of the 2a-NPA values of B-H vertexes are positive, and their functionalization is more challenging. The synthesis and full characterization of dibranched 9,10-R_2_-1,7-*closo*-carborane (R = CH_2_CHCH_2_, HO(CH_2_)_3_, Cl(CH_2_)_3_, TsO(CH_2_)_3_, C_6_H_5_COO(CH_2_)_3_, C_6_H_5_COO(CH_2_)_3_, N_3_(CH_2_)_3_, CH_3_CHCH, and C_6_H_5_C_2_N_3_(CH_2_)_3_) compounds as well as the tetrabranched 9,10-R_2_-1,7-R_2_-*closo*-C_2_B_10_H_8_ (R = CH_2_CHCH_2_, HO(CH_2_)_3_) are presented. The X-ray diffraction of 9,10-(HO(CH_2_)_3_)_2_-1,7-*closo*-C_2_B_10_H_10_ and 9,10-(CH_3_CHCH)_2_-1,7-*closo*-C_2_B_10_H_10_, as well as their Hirshfeld surface analysis and decomposed fingerprint plots, are described. These new reported tetrabranched *m*-carborane derivatives provide a sort of novel core for the synthesis of 3D radially grown periphery-decorated macromolecules that are different to the 2D radially grown core of the tetrabranched *o*-carborane framework.

## 1. Introduction

Icosahedral carborane clusters with empirical formula C_2_B_10_H_12_ can be in three different isomers: 1,2-*closo*-C_2_B_10_H_12_ (*o*-carborane), 1,7-*closo*-C_2_B_10_H_12_ (*m*-carborane; **1**), and 1,12-*closo*-C_2_B_10_H_12_ (*p*-carborane). Figure 1 displays a schematic representation of the isomers with their vertexes numbering. Despite their common icosahedral geometry, they display similarities, but also important differences. Among the similarities are the high stabilities and 3D geometrical properties, their very similar 3D aromatic character [1,2] that leads to display great inertia to keep the original scaffold upon electrophilic substitution, their dual-mode as electron-withdrawing through carbon or electron-donating through boron vertexes [3,4,5], their molecular volume that is high compared to rotating benzene [6], and high hydrophobicity [7,8,9]. Among the differences are the dipolar moment and their different reactivity towards boron elimination [8], and the lowest unoccupied molecular orbital (LUMO) geometrical disposition that is responsible for many of the physical properties of the isomers. Among the three of them, the most extensively studied is the *o*-carborane. Some tips to keep in mind between the three isomers when substitution is sought are: First, the weak acidic C_c_-H bond (C_c_ = cluster C atom) [10] can be deprotonated using a strong base followed by an electrophilic reaction to form the C_c_-R bond [8,11]. Second, the B-H hydrogen atoms with the hydridic character on (B(9,12), B(8,10), and B(4,5,7,11)) are subjected to electrophilic substitution to form B-halogen units [8,11] that, if followed by a Kumada cross-coupling reaction, may lead to the introduction of organic moieties to make B-R vertexes [3,4,5,6,7,8]. This procedure does not proceed equally for all B-Hs, for instance, at the B(3,6) vertices, which are the most electron-deficient vertexes, their functionalization does not take place by using the same process at the other cluster’s vertices [8,11]. Substitution on these positions can be achieved via the deboration-capping multistep reaction or via the metallation procedure [11,12,13]. The diverse regioreactivity of *o*-carborane, has been exploited and adapted to make *o*-carborane an exceptional core for developing a large variety of multibranched molecules, globular macromolecules, dendrimers (Figure 2b), and so on [14,15,16,17,18]. Moreover, new versatile synthons have been explored through the multi-functionalization of B and C_c_ atoms jointly, which make the *o*-carborane clusters an exciting platform for new materials [6,8,18,19,20,21,22,23,24,25].

On the other hand, the reactivity of *m*-carborane is less studied but for the C_c_-H vertexes, which are less acidic as compared to the C_c_-H vertexes of the *o*-isomer [10,26]. Using a similar strategy as for *o*-carborane, a wide variety of 1-R-1,7-*closo*-C_2_B_10_H_11_ and 1,7-R_2_-1,7-*closo*-C_2_B_10_H_10_ derivatives has been developed [27,28,29,30,31,32,33,34,35]. To some extent, the current state of knowledge of the *m*-carborane functionalization through the B-H vertexes is in an odd situation. As compared to the *o*-carborane, a much-limited number of protocols leading to modify the B-H vertexes in the *m*-cluster have been reported [8,11]. In this context, very few derivatives of *m*-carborane with a functional group that is bonded to B(9) or B(9) and B(10) synchronously have been described [27,28,29,30,31,32,33,34,35]. The Pd-catalyzed cross-coupling reaction of 9-X-1,7-*closo*-C_2_B_10_H_11_ and 9,10-X_2_-1,7-*closo*-C_2_B_10_H_10_ (X= halogen atom) represents one of these examples of derivatization [34,36,37]. By contrast to the *o*-carborane, no multibranched *m*-carborane structures with a general formula 1,7-R_2_-9,10-R’_2_-1,7-*closo*-C_2_B_10_H_8_ have been reported despite the potential of its structure and the relatively high reactivity of the C_c_-H bonds that should allow the reaction to a great extent. Notably, as shown in Figure 2, the *m*-carborane core provides a 3D radially growth core while *o*-carborane a 2D one.

Consequently, we became interested in introducing organic branches connected to B(9) and B(10) to prepare a new set of 9,10-R_2_-1,7-*closo*-C_2_B_10_H_10_ derivatives. In the second part of this paper, we functionalized the two C_c_-H of 9,10-(CH_2_=CHCH_2_)_2_-1,7-*closo*-C_2_B_10_H_10_ to form the quadruped-shaped structure with a general formula 1,7-R_2_-9,10-R’_2_-1,7-*closo*-C_2_B_10_H_10_ which might serve as versatile precursors with free ends for further reaction.

## 2. Results and Discussion

### 2.1. Synthesis of di-Branched m-carborane Derivatives at the 9,10 Vertexes

Versatile strategy for the synthesis of the two branches B(9,10) *m*-carborane derivatives (9,10-R_2_-1,7-*closo*-C_2_B_10_H_10_) was achieved by using 9,10-I_2_-1,7-*closo*-C_2_B_10_H_10_ as the starting compound.

The synthesis of 9,10-I_2_-1,7-*closo*-C_2_B_10_H_10_; (**2**) has been reported by using two different methodologies: i) the electrophilic iodination reaction of icosahedral *closo m*-carborane (**1**) by using a molar Equiv. Of iodine: monochloride, which acts as an electrophilic agent, in the presence of catalytic amounts of aluminum chloride, and ii) using iodine as an electrophilic agent in a very acidic media (HNO_3_:H_2_SO_4_, 1:1). The target compound **2** was obtained in 60% and 87% yield, respectively [38,39]. Our study focused on the synthesis of new Boron disubstituted *closo m*-carborane derivatives at the 9 and 10 vertexes began with the synthesis of **2** in 87% yield by combining the two reported methods: an equimolar ratio of *m*-carborane (**1**): iodine in acidic HNO_3_:H_2_SO_4_ (1:1) solution was left under reflux to react for 3 h (Scheme 1).

To produce the B-C bonds on **1**, a useful and general method is the Kumada cross-coupling reaction on B-iodinated *m*-carborane **2** with Grignard reagents in the presence of Pd(II) and Cu(I) catalysts. To achieve the di-branched *m*-carborane derivatives at the 9, 10 vertexes, the cross-coupling reaction on **2** was studied using CH_2_CHCH_2_MgCl Grignard derivative in the presence of [PdCl_2_(PPh_3_)_2_] and CuI as catalysts to give the 9,10-(CH_2_CHCH_2_)_2_-1,7-*closo*-C_2_B_10_H_10_ (**3**) in 95% yield [40].

The terminal olefin groups in **3** are ready for further reactions on them, enabling the *m*-carborane cluster to become the template for a new type of macromolecules having a rigid head and two appended branches. As a first example of these molecules, compound **3** was converted to 9,10-(HOCH_2_CH_2_CH_2_)_2_-*closo*-1,7-C_2_B_10_H_10_, (**4**) following the hydroboration/oxidation reaction on **3** by using BH_3_·THF as hydroboration agent and subsequent oxidation with H_2_O_2_ in a basic aqueous solution. After workup, crystalline white pure solid, **4**, was obtained in 93% yield. The ^1^H NMR spectrum displays a new broad peak at 3.43 ppm, which supports the presence of the O-H group in **4**. Also, the ^1^H and ^13^C{^1^H} NMR spectra revealed that the reaction had proceeded by an anti-Markovnikov addition, therefore having the two hydroxyl groups at terminal positions. No hindered hydroboranes were thus needed for the control of the reaction’s regioselectivity.

These terminal alcohol groups anticipate versatile chemistry for radial growth, given the availability of the terminal hydroxyl groups for further elongation of the chains. Moreover, the C_c_–H vertexes on the rigid *m*-carborane head are ready for derivatization or supramolecular assembly. Then, the *m*-carborane cluster, as *o*-carborane does, provides a singular platform for the construction of highly dense multibranched molecules with a wide range of possibilities. Therefore, derivatives of *m*-carborane with precise patterns of substitution, which are sterically different from the ones of *o*-carborane but complementary, can be prepared by a judicious choice of the synthetic procedure. Consequently, the substitution of the terminal hydroxyl units in **4** by chloro (**5**), ester (**6**), tosyl (**7**) or azide (**8**) groups, which enable the branches to grow by a subsequent coupling reaction with nucleophilic agents, was achieved (Scheme 2).

Chlorination in **3** was achieved using SOCl_2_ and [Nbu_4_]Cl salt to give 9,10-(ClCH_2_CHCH_2_)_2_-1,7-*closo*-C_2_B_10_H_10_ (**5**) in 92% yield. 

As an example of the esterification of the terminal alcohol groups, compound **6** was obtained in 90% yield by Steglich esterification [41,42] with benzoic acid using *N,N’*-dicyclohexylcarbodiimide as a coupling reagent and the *N,N*-dimethylaminopyridine as a catalyst.

Furthermore, alcohol groups were converted to tosylate groups by performing the reaction of **3** with tosyl chloride, Net_3_ as a base, and [HNMe_3_]Cl as a catalyst [43], obtaining 9,10-(TsOCH_2_CH_2_CH_2_)_2_-1,7-*closo*-C_2_B_10_H_10_, **7**, in 85% yield.

Overall, we have succeeded in the preparation of these new *m*-carborane derivatives **5**–**7** thanks to the primary alcohol groups, which undergo chain extension reactions. Owing to the formation of **5**, a new way of functionalization is opened to prepare the azide derivative **8**, which in turn opens the way to perform the Azide-Alkyne Huisgen Cycloaddition commonly known as the click reaction. Compound **8** was obtained in 81% yield by avigorous stirring of **5**, with excess of NaN_3_ and [Nbu_4_]Cl in a mixture of toluene and water, at reflux for 24–48 h. An example of the click reaction on **8** was the compound **9** synthesis in 86% yield by simple reaction with phenylaceylene, sodium ascorbate, and hydrated CuSO_4_ as a catalyst in a mixture of dioxane/water.

Therefore, di-branched *m*-carborane derivatives (Scheme 2) with precise patterns of substitution are prepared by judicious choice of the synthetic procedure using similar conditions to the preparation of *o*-carborane derivatives present in previous work [14].

### 2.2. Synthesis of tetra-Branched m-carborane Derivatives at the 1,7,9,10 Vertexes

The above described di-branched 9,10-(CH_2_=CHCH_2_)_2_-1,7-*closo*-C_2_B_10_H_10_ (**3**) derivative, which still possesses its two C_c_-H vertexes are ready for derivatization, offers the possibility to obtain globular icosahedral *m*-carborane derivatives with four branches as a new dendritic structure (Figure 2) by the incorporation of functional groups at the two carbon vertexes. Consequently, starting with **3** the four branched 1,7-(CH_2_=CHCH_2_)_2_-9,10-(CH_2_=CHCH_2_)_2_-1,7-*closo*-C_2_B_10_H_8,_
**10**, is obtained in two steps: i) removing the acidic hydrogen atoms with two equivalents of BuLi and ii) by electrophilic reaction with two equivalents of allylbromide (Scheme 3). From **10**, the tetraalcohol 1,7-(OHCH_2_CH_2_CH_2_)_2_-9,10-(OHCH_2_CH_2_CH_2_)_2_-1,7-*closo*-C_2_B_10_H_8_**, 11**, can be achieved by hydroboration. In the same way, **11** is ready as a core for constructing a tetra-branched *m*-derivatives using the judicious choice of synthetic procedure (Scheme 3). 

The synthesis of the tetra-substituted dianionic compound [9,10-({3,3’-Co(8’-(OCH_2_CH_2_)_2_-1’,2’-C_2_B_9_H_10_)(1’’,2’’-C_2_B_9_H_11_)}_2_-1,7-C_2_B_10_H_8_)^2−^ was attempted in THF starting with compound **3** in a two steps reaction: i) the deprotonation of C_c_-H vertexes of **3** using two equivalents of *n*-BuLi to form the intermediate Li_2_[9,10-(CH_2_=CHCH_2_)_2_-1,7-*closo*-C_2_B_10_H_8_]] salt and, ii) the nucleophilic attack of this salt to the dioxanate ring of the zwitterion [3,3’-Co-(8-(CH_2_CH_2_O)_2_-1,2-C_2_B_9_H_10_)(1’,2’-C_2_B_9_H_11_)] in the same way as Li_2_[1,7-*closo*-C_2_B_10_H_10_]] had performed (see Scheme 4) [44]. Nevertheless, unexpectedly, the synthesis of the dianionic compound was not achieved while the isomerization of allyl branches to propenyl ones took place giving the isomer 9,10-(CH_3_CH=CH)_2_-1,7-*closo*-C_2_B_10_H_10_, **12**, in 80% yield (Scheme 4).

The reason for this unexpected reaction can be the comparable acidity of the allyl groups and the C_c_-H of the *m*-carborane unit, which may allow a deprotonation/protonation isomerization of the allyl group as it is well known for allylbenzenes [45]. The pKa value of the unsubstituted carborane clusters, which are insoluble in water, have been determined by two methods [6,10]. The pKa by using Streitwieser’s scale provides the 27.9 value for the isomers *m*- carborane, while the one obtained by polarography is 24 [46]. Both experimental techniques support that unsubstituted *m*-carborane is a very weak Brønsted acid [46]. The allyl isomerization of **3** to propenyl in **12**, which takes place in THF, is supported by the formation of solvent separated ion pairs that prevent the carboranyl anion to act as a nucleophile. To verify this hypothesis, Density-functional theory (DFT) calculations were performed (details in the S.I.). The proton affinity (PA) of the cluster carbon atom is 332.8 kcal/mol (at B3LYP-D3/6-311+G**, PCM=tetrahydrofuran level of theory), while the proton affinity of allylic carbon atom has a somewhat higher value (342.3 kcal/mol). This moderate difference (∆PA = 9.5 kcal/mol) probably allows for the above-mentioned mechanism. The question arises whether the same process does not occur in the case of the analog *o*-carborane based compounds [44]. It is known that cluster carbon in *m*-carborane is more than 1000 times less acidic than its *orto* isomer [47,48] therefore the difference between the two positions (allylic *vs* carboranyl) is larger as it was verified by our calculations (∆PA = 18.6 kcal/mol) as well. It should be highlighted that Li^+^ mediated isomerizations on allyl substituents bonded at the C_c_ vertexes of the *o*-carborane cluster was previously demonstrated as well, as Et_2_O does not tend to induce isomerization, whereas THF or DME produces the propenyl isomer [49]. A similar mechanism should be considered as well.

^1^H-NMR spectrum of **12** supported the allyl branches isomerization to propenyl ones but, this process was unambiguously proven by X-ray diffraction of **12** from good crystals, which were grown from its acetone solution.

### 2.3. Characterization of di-Branched m-carborane Derivatives at the 9,10 Vertexes

The electrophilic substitution of the *o*-carborane led to the formation of the tetrasubstituted 8,9,10,12-I_4_-1,2-*closo*-C_2_B_10_H_8_ compound [5,42,43] in which the B-I vertexes reside at the compacted adjacent positions antipodal to the two cluster carbon C_c_ atoms. Conversely, the iodination electrophilic substitution takes place only at the B(9) and B(10) vertexes of the *m*-isomer.

We reported that the 2a-NPA value for a bond, defined as the sum of the NPA charges of the two bonded atoms (e.g., B-H or C-H), matches the order of attack on the different cluster’ bonds [50]. Calculated NPA charges of the two bonded atoms (2a-NPA, calculated at B3LYP-D3/6-311+G** level of theory) of *ortho*- and *meta-**closo*-carborane (present in Table 1) explain the higher accessibility of the B-H vertexes of *o*-carborane cluster to undergo electrophilic reaction, which drive to the formation of B-I bonds. While in the case of *o*-carborane there are four negative 2a-NPA values, in the case of *m*-carborane, there are only two. However, these positions exhibit higher reactivity towards electrophilic agents. Since in the case of *m*-carborane most of the 2a-NPA values of B-H vertexes are positive, the functionalization of this compound is more challenging. Table 1 shows that the electron density at the B-H vertexes of *m*-carborane follow a different trend B(9), B(10) >>> B(5), B(12) > B(4), B(6), B(8), B(11) B(2), B(3) than *o*-carborane, which is B(9,12) > B(8,10) > B(4,5,7,11) > B(3,6) [3,51]. Contrary to the *o*-carborane that contains two positive natural charges; the *m*-carborane presents four positive natural charges on BH vertex, which explain the difficulty of the substitution of the B-H vertexes. Figure 3 shows that LUMO in o-carborane is located between the C atoms, whereas it is not the case for m-carborane where it is more disperse. Therefore, the carbon cluster position in the carborane has an important role related to the substitution of the B-H vertexes. Using the electrophilic iodination, it is possible to derivatize only B(9) and B(10) because these boron atoms do not have any connection with the C_c_ in the *meta* isomer. On the contrary, for the *o*- isomer the same procedure allows the attack to all B-H vertexes except B(3) and B(6) that are adjacent to both carbon clusters [52].

The starting (**2** and **3**) and new compounds (**4**–**9**) were fully characterized by ^1^H, ^1^H{^11^B}, ^11^B, ^11^B{^1^H}, ^13^C{^1^H}, and 2D COSY ^11^B {^1^H}-^11^B {^1^H} NMR spectroscopic techniques to be taken as inputs for the discussion of the influence of the substituents at the 9,10 vertexes on the Boron disubstituted *closo m*-carborane derivatives.

The ^11^B{^1^H} NMR spectrum of the parent cluster **1** displays four signals with intensities 2:2:4:2 from low to high field −5.6, −9.5, −12,0 and −15.4 ppm, which corresponds to a weighted average ^11^B{^1^H} NMR chemical shift, <δ(^11^B)> ≈ −10.9 ppm [53]. Conversely, the ^11^B {^1^H} NMR spectrum of **2** displays four signals with intensities 2:4:2:2 from low to high field at –3.0, –10.4, –17.0, and –19.5 ppm, which corresponds to a weighted average <δ(^11^B)> ≈ −12 ppm. The presence of the two iodo groups bonded to the B(9,10) in **2** produces a <δ(^11^B)> upfield of −1.1 ppm in the ^11^B NMR. The upfield resonance of the ^11^B {^1^H} NMR spectrum of **2** at −19.5 ppm does not split into a doublet in the ^11^B-NMR spectrum supporting that it corresponds to the B-I at the 9 and 10 vertexes.

^11^B {^1^H}-^11^B {^1^H} 2D COSY NMR is of enormous use and potential in polyhedral boron chemistry because it provides a way of rapidly assigning ^1l^B resonances [54,55]. To assign the resonances of compounds **1** and **2** to the different cluster’s vertexes by NMR spectroscopy, the two-dimensional ^11^B {^1^H}-^11^B {^1^H} COSY NMR spectra of compounds **1** and **2** were run (See Appendix A). Once the B(9,10) has been unambiguously assigned in compounds **1** and **2**, it is possible to confirm that the substitution of hydrogen by iodo causes significant shielding (−10 ppm) on the boron atoms that support the iodo units. ^11^B {^1^H}-^11^B {^1^H} 2D COSY NMR spectra of compounds **1** and **2** allow assigning the vertexes’ resonances for **1** and **2** (see Appendix A). On the other hand, the assignment of the hydrogen atoms to the respective boron cluster vertexes was done by running the selective irradiation ^1^H{^11^B} NMR spectra (Table 2) which confirms the presence of four signals in **1** and only three in **2**. Notably, all proton resonances were shifted upfield in the ^1^H{^11^B} NMR spectrum which demonstrates the influence of the Iodo groups on all clusters’ vertexes. 

The ^11^B{^1^H} NMR spectrum of **3** displays four signals with intensities 2:2:4:2 from low to high field at 0.6, −5.4, −12.5, and −19.1 ppm, which corresponds to a weighted average <δ(^11^B)> of ca. −9.8 ppm. The peak at +0.6 ppm does not split into a doublet in the ^11^B-NMR spectrum, which supports the substitution of an iodo by carbon from the allyl group, which causes a downfield shift on the boron vertexes. On the other hand, the ^1^H and ^1^H{^11^B} NMR spectra are useful to identify the presence of the organic fragments linked to the carborane cluster. Figure 4a shows the presence of three new signals (area ratio 1:2:2, from low to the high field), which are related to the typical resonances of terminal allyl groups. The protons bonded to the boron vertex (H_d_) appear as a doublet (^1^*J*(H,H) = 7.7 Hz) at 1.78 ppm. There is an overlap of H_a_ and H_b_ resonances, which should appear, each one, as a double doublet, but looks like a triplet at 4.88 ppm. H_c_ is the most complicated proton of the allyl group because of the presence of four different protons at its neighboring carbon atoms. This appears in the range 5.97–5.82 ppm as a multiplet. The coupling constants of H_c_ with neighbors is shown in Figure 4b.

Compounds **4, 5**, **6,** and **7** were also characterized by ^1^H, ^1^H{^11^B}, ^11^B, ^11^B{^1^H} and ^13^C{^1^H} NMR spectroscopy.

Table 3 lists the ^11^B{^1^H} NMR chemical shifts for the B(9,10) disubstituted *m*-carborane derivatives while Table 4 summarizes the ^1^H, ^13^C{^1^H } NMR spectra and the stretching frequency of C_c_-H in the IR spectra for the B(9,10) disubstituted *m*-carborane derivatives. The presence of organic branches connected to B(9) and B(10) causes a resonance downfield shift about +11 ppm on these boron atoms. Therefore, the average chemical shift value <δ(^11^B)> = –10.9 ppm of parent **1** is around –9.6 ppm for 9,10-R_2_-1,7-*closo*-C_2_B_10_H_12_ derivatives (R=CH₂=CH–CH₂, HO(CH_2_)_3_, Cl(CH_2_)_3_, PhCOO(CH_2_)_3_, CH_3_-C_6_H_4_-SO_3_(CH_2_)_3_). There is no difference in these two features between the two isomers, *ortho* and *meta*.

Table 3 shows a downfield shift (Δδ = +10.1 ppm) of the B(9,10) resonances of 9,10-(allyl)_2_-1,7-*closo-*C_2_B_10_H_10_ (**3**) *vs* the corresponding B(9,10)-H ones in the parent *m*-carborane. A similar downfield (Δδ = +10.6 ppm) is reported for the B(9,12) vertexes of 9,12-(allyl)_2_-1,2-*closo*-C_2_B_10_H_10_ with respect to the B(9,12)-H vertexes of the parent *o*-carborane [14]. The ^11^B{^1^H} NMR spectrum provides information on the electron density surrounding B atoms in the cluster vertexes, so it can be concluded that the effect of a B-allyl vertex concerning to the former B-H in the ^11^B{^1^H} NMR of both isomers is almost the same, Δδ +10.6 ppm and +10.1 ppm, for *o*- and *m*-, respectively. However, there is a major difference in the chemical shifts of the B-allyl nuclei of the two isomers: δ = +7.75 pm for 9,12-(allyl)_2_-1,2-*closo-*C_2_B_10_H_10_ and δ = +0.6 pm for 9,10-(allyl)_2_-1,7-*closo-*C_2_B_10_H_10_. We should remember that B-allyl vertexes are located antipodal to the C_c_ vertexes in the *o*- isomer but antipodal to B vertexes in the *m*-isomer. This fact indicates a quite relevant different electronic surrounding in the B-allyl sites in both isomers, which depends on the atoms’ nature at the antipodal vertexes.

Table 4 summarizes the ^1^H and ^13^C{^1^H } NMR spectra and stretching frequencies of C_c_-H bonds in the IR spectra for the reported 9,10-R_2_-1,7-*closo*-C_2_B_10_H_10_ derivatives; the presence of the allyl branches at the B(9,10) vertexes produces an upfield of the carbon and hydrogen atoms resonances of the C_c_-H concerning to the parent *m*-carborane in their ^1^H and ^13^C{^1^H } NMR spectra.

In Table 5, the comparison of the influence of the substituents at the B(9,12) in the *o*-carborane and the B(9,10) in the *m*-carborane is listed. To notice is that the influence on the chemical shift of the B-halogen (halogen = Cl, Br, I) vertexes in both isomers is the same: iodo is larger than bromo and bromo is larger than chloro. This is due to the i) electronegativity of halogen atoms, which follows the trend Cl > Br > I and ii) π back donation of halogen is I > Br > Cl.

### 2.4. Characterization of Tetrabranched m-carborane Derivatives at the 1,7,9,10 Vertexes

From the analysis of the ^1^H NMR spectra of **10**, it is seen that the original signal corresponding to the protons linked to the carbon cluster, which appear at 2.83 ppm, vanishes while new signals at 5.58, 5.00 and 2.56 ppm corresponding to the allyl branches on these C_c_ vertexes are distinguished.

An important influence of the presence of the organic branches linked to the two C_c_ atoms is observed in the ^11^B downfield shift of the B(2) and B(3) vertexes that move from –19.1 ppm in **3** to -15.7 ppm in **10**. To notice is the upfield shift of <δ(^11^B)> when moving from **2** to **10**: <δ(^11^B)> = –12.0 ppm in compound **2** (with two B-I and two C_c_-H vertexes), <δ(^11^B)> = -10.9 ppm on the parent *m*-carborane, <δ(^11^B)> = –9.8 ppm in **3** (with two B-allyl and two C_c_-H vertexes), <δ(^11^B)> = –8.2 ppm in **10** (with two B-allyl and two C_c_-allyl vertexes) (see Table 3). Consequently, the incorporation of organic branches at the cluster vertexes produces a downfield of <δ(^11^B)> in the ^11^B NMR while the iodo groups have the opposite effect, supporting that cluster-only total charge is dissimilarly affected by electron-withdrawing substituents than electron-donating ones.

### 2.5. Structural Description

#### 2.5.1. Crystallographic Studies

A search in the Cambridge Structural Database [57] showed just 3 hits (CUWMUD, TOKCUR and YOZSOV) for 9,10-R_2_-1,7-*closo*-C_2_B_10_H_10_ for R = -CCH, -CH_2_C_6_H_4,_ and -C_6_H_5_, respectively [34,36,37]. In this paper, we contribute with two additional X-ray structures that provide a broader view of the *m*-carborane derivatives.

To get information in such a family of compounds, good crystals of 9,10-(HOCH_2_CH_2_CH_2_)_2_-1,7-*closo*-C_2_B_10_H_10_ (**4**) and 9,10-(CH_3_CH=CH)_2_-1,7-*closo*-C_2_B_10_H_10_ (**12**) suitable for X-ray- diffraction were grown from an acetone solution at low temperature. Compound **4** was solved in the triclinic system, with a P 1 space group with four molecules in the asymmetric unit (Z = 4) and all atoms laid on the 1(a) Wyckoff positions. Compound (**12**) also solved in the triclinic system, but in a different space group (P-1) with two molecules in the asymmetric unit (Z = 2) and all atoms laid in 1(i) Wyckoff position. Figure 5 shows the crystal structures of **4** and **12** with the corresponding atom labels. Table 6 displays all crystallographic data and selected bond distances and angles are in the Appendix A.

Compound **4** is the first example of a B(9,10) disubstituted *closo* 1,7-carborane derivative with a terminal O-H group. Furthermore, compound **12** is the first example of *closo m*-carborane with branches containing double bonds. For this, the behaviour of the two branches in the crystal network has been studied in detail. Exploring the crystal self-assembly, the presence of H^…^H short contacts in the range from 1.207 Å to 2.240 Å for compound **4** and equal to 2.252 Å for compound **12** are noticed, which are presented in Figure 6 and Figure 7, respectively. In carborane chemistry, the dihydrogen H^…^H short contacts are related to the presence of two types of H atoms: the acidic C_c_-H and the hydride B-H [58]. The supramolecular structure of **4** has an extensive network of hydrogen bonding due to the presence of the terminal OH groups (Figure 6a,b) with intermolecular distances shorter that sum of Van Der Waals radii (∑vdW) minus 0.8 Å [59] and O-H···O angle values of 163.1° and 177.9°. The crystal packing of **4** is also stabilized by O^…^O as shown in Figure 6c. Accordingly, three different types of H^…^H short contacts were observed for **4**: C7-H7^…^H16-O16, O20-H20^…^H16-O16, and B3-H3^…^H14-C14. 

As expected, the presence of double bonds in **12** has a noticeable role in the stabilization of the supramolecular network (Figure 7). The π electronic effect of the double bond leads to the formation of the π^…^H-C_c_ contacts (brown dashed lines), which are substantially shorter than 2.90 Å corresponding to the sum of the van der Waals radii (∑vdW) [60]. The layers of **12** are connected into the final 3D structure through the B3-H3^…^H15A-C15 bonds due to the acceptor character of the hydrogen-bonded to the boron and the donor character of the hydrogen atoms of the -CH_3_ group.

#### 2.5.2. Hirshfeld Surface Analysis

The Hirshfeld surface analysis, which is a very valuable method for the analysis of intermolecular contacts that offers a whole-of-the-molecule approach [61], presents three different colours to study the intermolecular interactions in crystal structures. The red colour means the presence of an intermolecular distance shorter that ∑vdW, white colour indicates the presence of intermolecular distances close to ∑vdW and blue colour designates the contacts longer than ∑vdW. Moreover, the shape index on the Hirshfeld surface identify hollows (with shape index < 0) and bumps (with shape index > 0), which are related to the character of each atom; the presence of an acceptor atom is marked by a concavity and the presence of a donor one is marked by a convexity. Therefore, the previous results were corroborated by studying the Hirshfeld surface of both structures using the crystal explorer program [62]. In this respect, Figure 8 presents the d_norm_ of **4** and **12** to visualize the intermolecular interactions and their contribution towards the supramolecular network. The two-dimensional fingerprint plots, which provide information about the percentage of intermolecular contacts present in the Hirshfeld surface is present in the S.I. 

The darkest red area in the d_norm_ surface of **4** is observed at the end of the molecule, arising from the O^…^H short contact as presented in Figure 8 and confirmed in the fingerprint plots (See Appendix A). Furthermore, the d_norm_ surface has shown the presence of bright red areas related to the presence of the O^…^O and H^…^H short contacts. Because of this packing arrangement, the O atoms at the molecular extremity present an important behaviour on the stability of this molecule by showing close contact values with H atoms of adjacent molecules shorter than ∑vdW.

The d_norm_ presentation of compound **12** (Figure 8 and S.I.) shows the presence of dark red points related to the classic H^…^H bonds. The existence of π^…^H short contacts is observed as a bright red point. The presence of the π acceptor interactions is indicated by the appearance of red concave triangles surrounded by blue ones in the shape index surface (Figure 9a) [63] while the C_c_-H donor is confirmed by the blue convex area (Figure 9b) [64].

Despite the presence of many strong intermolecular interactions with contacts shorter than the sum of the van der Waals radii minus 0.80 Å, the H^…^H interactions are the most dominant with 87.8% and 94.2% in compounds **4** and **10** respectively, as shown in the fingerprint plots (See S.I.). The presence of 10.2% contacts O^…^O in **4** and 5.8% of π^…^H short contacts in **12** are also relevant.

## 3. Materials and Methods

### 3.1. Experimental Section 

Materials and instrumentation: All *m*-carborane clusters prepared are air-stable. All manipulations were carried out under nitrogen atmosphere. THF and DMF were distilled from sodium benzophenone before use. Reagents were obtained commercially and used as purchased without purification. 1,7-*closo*-C_2_B_10_H_12_ was obtained from Katchem.

ATR-IR spectra (ν, cm^−1^) were obtained using the a JASCO FT/IR-4700 spectrometer on a high-resolution (Madrid, Spain). The ^1^H and ^1^H{^11^B} NMR (300.13 MHz), ^13^C{^1^H} NMR (75.47 MHz), and ^11^B and ^11^B{^1^H} NMR (96.29 MHz) spectra were recorded on a Bruker ARX300 instrument equipped with the appropriate decoupling accessories (Bruker Biospin, Rheinstetten, Germany)). All NMR spectra were performed in the indicated deuterated solvent at 22 °C. The ^11^B and ^11^B{^1^H} NMR chemical shifts were referenced to external BF_3_·OEt_2_, while the ^1^H, ^1^H{^11^B}, and ^13^C{^1^H} NMR shifts were referenced to SiMe_4_. Chemical shifts are reported in units of parts per million downfield from reference, and all coupling constants in Hz.

#### 3.1.1. Synthesis and Characterization of **3**

The procedure for the synthesis of **3** was similar to that previously reported [40]. To a stirred solution of 9,10-I_2_-1,7-*closo*-C_2_B_10_H_10_
**2**, (300 mg, 1.34 mmol) in THF (15 mL) cooled to 0 °C in an ice-water bath was added, drop wise, a solution of allylmagnesium chloride in THF (6.06 mL, 1 M, 6.06 mmol). After stirring at room temperature for 30 min, [PdCl_2_(PPh_3_)_2_] (21.28 mg, 4% equiv.) and CuI (5.77 mg, 4% equiv.) were added in a single portion, following which the reaction was heated to reflux overnight. The solvent was removed, and 20 mL of diethyl ether were added to the residue. The excess of Grignard reagent was destroyed by slow addition of dilute HCl. The organic layer was separated from the mixture, and the aqueous layer was extracted with diethyl ether (3 × 10 mL). The combined organic phase was dried over MgSO_4_, filtered and the solvent removed under reduced pressure. The crude product was dissolved in hexane/chloroform mixture (1:1 by volume, ca. 5 mL) and passed rapidly through a bed of silica. The solvent was removed in a vacuum to give 9,10-(CH_2_=CHCH_2_)_2_-1,7-*closo*-C_2_B_10_H_10_
**3** as a yellowish oil (161.2 mg, 95%). Elemental analysis: calc: %C 42.8, %H 8.7; exp: %C 45.9, %H 8.7. ATR: ν = 3062 (vs, (C_c_-H and =CH_2_)), 2972, 2902 (vs, (=CH- and -CH_2_-)), 2592 (vs, (B-H)), 1634 (vs, (C=C)), 995, 978, 692 (s, (=CH)). ^13^C{^1^H} NMR (75.47 MHz, CDCl_3_) δ: 139.96 (s, CH_2_=*C*HCH_2_), 112.09 (s, *C*H_2_=CHCH_2_), 52.35 (s, *C*_c_-H), 21.67 (m, CH_2_=CH*C*H_2_). ^1^H-NMR (300.13 MHz, CDCl_3_) δ: 5.89 (m, 2H, CH_2_=C*H*CH_2_), 4.88 (m, 4H, C*H*_2_=CHCH_2_), 4.94 (s, 2H, C_c_-*H*), 1.78 (m, 4H, CH_2_=CHC*H*_2_). ^1^H{^11^B} NMR (300.13 MHz, CDCl_3_) δ: 5.89 (m, 2H, CH_2_=C*H*CH_2_), 4.88 (m, 4H, C*H*_2_=CHCH_2_), 4.94 (s, 2H, C_c_-*H*), 2.50 (s, 2H, B(5,12)-*H*), −2.24 (s, 2H, B(2,3)-*H*), 2.14 (s, 4H, B(4,6,8,11)-*H*), 1.77 (d,^3^*J*(H,H) = 7.8, 4H, CH_2_=CHC*H*_2_). ^11^B NMR (96.29 MHz, CDCl_3_) δ: −0.1 (s, 2B, B(9,10)), −6.2 (d, ^1^*J*(B,H) = 160, 2B, B(5,12)), −13.5 (d, ^1^*J*(B,H) = 163, 4B, B(4,6,8,11)), 20.3 (d, ^1^*J*(B,H) = 180, B(2,3)).

#### 3.1.2. Synthesis and Characterization of **4**

To a stirred solution of 9,10-(CH_2_=CHCH_2_)_2_-1,7-*closo*-C_2_B_10_H_10_, **3**, (150 mg, 0.67 mmol) in THF (2 mL) at 0 °C, was added, drop wise, a solution of BH_3_·THF in THF (1.37 mL, 1 M, 1.37 mmol). The resulting suspension was stirred at 0 °C for 30 min and at room temperature for further 30 min. Then, the reaction mixture was cooled again to 0 °C in an ice-water bath and water (2 mL) was slowly added. When gas evolution had stopped, an aqueous KOH solution (1.68 mL, 3M, 1.94mmol) and subsequently, H_2_O_2_ in water (0.20 mL, 35%, 2.30 mmol), were added. Stirring was maintained at room temperature for 1.5 h, after which two liquid phases were observed. The upper organic layer was separated from the mixture and the aqueous layer and washed with THF (3 × 2 mL). The combined organic phase was dried over MgSO_4_, filtered and the solvent removed in vacuo to give 9,10-(HOCH_2_CH_2_CH_2_)_2_-1,7-*closo*-C_2_B_10_H_10_
**4**. Yield: 151 mg (87%). Elemental analysis: calc: %C 36.9, %H 9.22; exp: %C 36.79, %H 8.2. ATR: ν= 3305 (vs, ν_s_(O-H)), 3038 (vs, C_c_-H), 2930, 2886,2850, 2823 (vs, C_alkyl_-H), 2585 (s, B-H), 1055, 1005, 978 (s, C-O). ^13^C{^1^H} NMR (300.13 MHz, (CD_3_)_2_CO) δ: 64.20 (s, HO*C*H_2_CH_2_CH_2_), 52.95 (s, *C*_c_), 32.27 (s, HOCH_2_*C*H_2_CH_2_), 10.23 (s, HOCH_2_CH_2_*C*H_2_). ^1^H-NMR (300.13 MHz, (CD_3_)_2_CO) δ: 3.49 (s, 2H, C_c-_*H*), 3.55 (m, 4H, HOC*H*_2_CH_2_CH_2_), 3.40 (t, 2H, *H*OCH_2_CH_2_CH_2_), 1.60 (m, 4H, HOCH_2_C*H*_2_CH_2_), 0.81 (t, ^3^*J*(H,H) = 16.7, 4H, HOCH_2_CH_2_C*H*_2_). ^1^H{^11^B} NMR (300.13 MHz, (CD_3_)_2_CO) δ: 3.49 (s, 2H, C_c-_*H)*, 3.55 (m, 4H, HOC*H*_2_CH_2_CH_2_), 3.40 (t, 2H, *H*OCH_2_CH_2_CH_2_), 2.48,2.20,2.08 (s, 8H, B*-H*), 1.60 (m, 4H, HOCH_2_C*H*_2_CH_2_), 0.81 (t, ^3^*J*(H,H) = 16.7, 4H, HOCH_2_CH_2_C*H*_2_). ^11^B NMR (96.29 MHz, (CD_3_)_2_CO) δ: 1.9 (s, 2B, B(9,10)), −5.3 (d, ^1^*J*(B,H) = 157, 2B, B(5,12)), −12.7 (d, ^1^*J*(B,H) = 160, 4B, B(4,6,8,11)), −19.3 (d, ^1^*J*(B,H) = 178, 2B, B(2,3)). Colourless good crystals suitable for X-ray diffraction were grown in acetone.

#### 3.1.3. Synthesis and Characterization of **5**

To a stirred solution of 9,10-(HOCH_2_CH_2_CH_2_)_2_-1,7-*closo*-C_2_B_10_H_10_, **4**, (300 mg, 1.14 mmol) and [NBu_4_]Cl (132.59 mg, 0.478mmol) in dry THF (10mL) at 0 °C, was added SOCl_2_ dropwise (0.52 mL, 7.076 mmol). The resulting solution was stirred at 0 °C for 1 h and at room temperature overnight. The solvent was removed under reduced pressure, and 8 mL of diethyl ether were added. A solution of Na_2_CO_3_ (8 mL, 2 M) was slowly added with stirring. The mixture was thoroughly shaken, and the two layers separated. The aqueous layer was extracted with diethyl ether (3 × 5 mL). Then, the combined organic phase was separated and a solution of HCl (8 mL, 0.1 M) was added, the mixture was thoroughly shaken again. The upper organic layer was separated from the mixture, and the aqueous layer was washed with diethyl ether (3 × 5 mL). Finally, the combined organic phase was dried over MgSO_4_, filtered and the solvent removed in vacuo to give 9,10-(ClCH_2_CH_2_CH_2_)_2_-1,7-*closo*-C_2_B_10_H_10_, **5**. Yield: 310 mg (92%). Elemental analysis: calc: %C 32.32, %H 7.40; exp: %C 33.04, %H 7.60. ATR: ν= 3053 (s, C_c_-H), 2989, 2972, 2902 (s, C_alkyl_-H), 2626, 2587 (s, B-H), 1310, 1279 (s, CH_2_-Cl), 728, 647 (s, C-Cl). ^13^C{^1^H} NMR (300.13 MHz, (CD_3_)_2_CO) δ: 53.28 (s, *C*_c_), 47.14 (s, Cl*C*H_2_CH_2_CH_2_), 32.98 (s, ClCH_2_*C*H_2_CH_2_), 11.47 (s, ClCH_2_CH_2_*C*H_2_). ^1^H-NMR (300.13 MHz, (CD_3_)_2_CO) δ: 3.53 (s, 2H, C_c_-*H*), 3.61 (t, 4H, ClC*H*_2_CH_2_CH_2_), 1.87 (m, 4H, ClCH_2_C*H*_2_CH_2_), 0.94 (t, 4H, ClCH_2_C*H*_2_CH_2_). ^1^H{^11^B} NMR (300.13 MHz, (CD_3_)_2_CO) δ: 3.53 (s, 2H, C_c_-*H)*, 3.61 (t, 4H, ClCH_2_C*H*_2_CH_2_), 2.51, 2.23, 2.13 (s, 8H, B*-H*), 1.87 (m,4H, ClCH_2_C*H*_2_CH_2_), 0.78 (t, 4H, ClCH_2_CH_2_C*H*_2_). ^11^B NMR (96.29 MHz, (CD_3_)_2_CO) δ: 1.3 (s, 2B, B(9,10)), −5.3 (d, ^1^*J*(B,H) = 155, 2B, B(5,12)), −12.6 (d, ^1^*J*(B,H) = 159, 4B, B(4,6,8,11)), −19.0 (d, ^1^*J*(B,H) = 179, 2B, B(2,3)).

#### 3.1.4. Synthesis and Characterization of **6**

To a stirred solution of 9,10-(HOCH_2_CH_2_CH_2_)_2_-1,7-*closo*-C_2_B_10_H_10_ (94 mg, 0.361 mmol), **4**, 4-*N,N*-dimethylaminopyridine (97.17 mg, 0.795 mmol), *N,N’*-dicyclohexylcarbodiimide (164.11 mg, 0.795 mmol) and benzoic acid (97.17 mg, 0.795 mmol) in dry dichloromethane (10 mL). The resulting solution was stirred at room temperature for 1 h. The white precipitate (dicyclohexylurea) is filtered and then an extraction using 10 mL of HCl (1 M) was done. The aqueous layer was extracted with CH_2_Cl_2_ (3 × 10 mL). The combined organic phase was dried over MgSO_4_, filtered, and the solvent removed under reduced pressure to give 152 mg (90 %) of 9,10-(C_6_H_5_COOCH_2_CH_2_CH_2_)_2_-1,7-*closo*-C_2_B_10_H_10_, **6**. Elemental analysis: calc: %C 56.41, %H 6.83; exp: %C 55.72, %H 6.81. ATR: ν= 3064 (vs, C_c_-H), 2988, 2971, 2904 (vs, C_alkyl_-H), 2594, 2565 (s, B-H), 1711 (s, v_s_(C=O)), 1599 (s, C-O). ^13^C{^1^H} NMR (300.13 MHz, (CD_3_)_2_CO) δ: 165.87 (s, C_6_H_5_*C*OOCH_2_CH_2_CH_2_), 132.82 (s, C_aryl_), 130.70 (s, C_aryl_), 129.22 (s, C_aryl_), 128.45 (s, C_aryl_), 66.71(s, C_6_H_5_COO*C*H_2_CH_2_CH_2_), 53.27 (s, *C*_c_), 25.37 (s, C_6_H_5_COOCH_2_*C*H_2_CH_2_) 10.06 (s, C_6_H_5_COOCH_2_CH_2_*C*H_2_). ^1^H-NMR (300.13 MHz, (CD_3_)_2_CO) δ: 8.03 (d, ^1^*J*(H,H) = 8.1, 4H, H_aryl_), 7.65 (t, 2H, H_aryl_), 7.53 (m, 4H, _aryl_), 4.30 (t, 4H, ^3^*J*(H,H) = 6.8, C_6_H_5_*C*OOCH_2_CH_2_CH_2_), 3.55 (s, 2H, C_c_-*H*), 1.87 (m, 4H, C_6_H_5_*C*OOCH_2_CH_2_CH_2_), 0.98 (m, 4H, C_6_H_5_*C*OOCH_2_CH_2_CH_2_). ^1^H{^11^B} NMR (300.13 MHz, (CD_3_)_2_CO) δ: 8.03 (d, ^1^*J*(H,H)= 8.1, 4H, H_aryl_), 7.65 (t, 2H, H_aryl_), 7.51 (m, 4H, H_aryl_), 4.30 (t, ^3^*J*(H,H) = 6.8, 4H, C_6_H_5_*C*OOCH_2_CH_2_CH_2_), 3.55 (s, 2H, C_c_-H), 2.52, 2.27, 2.16 (s, 8H, B*-*H), 1.87 (m, 4H, C_6_H_5_COOCH_2_CH_2_CH_2_), 0.98 (m, 4H, C_6_H_5_COOCH_2_CH_2_CH_2_). ^11^B NMR (96.29 MHz, (CD_3_)_2_CO) δ: 1.6 (s, 2B, B(9,10)), −5.2 (d, ^1^*J*(B,H) = 149, 2B, B(5,12)), −12.5 (d, ^1^*J*(B,H) = 158, 4B, B(4,6,8,11)), −19.1 (d, ^1^*J*(B,H) = 175, 2B, B(2,3)).

#### 3.1.5. Synthesis and Characterization of **7**

To a mixture of 9,10-(HOCH_2_CH_2_CH_2_)_2_-1,7-*closo*-C_2_B_10_H_10_, **4**, (163 mg, 0.627 mmol) and [HNMe_3_]Cl (12.76 mg, 0.13 mmol) in 5 mL of dry toluene 1.5 mL of Triethylamine was added. In second flask, the p. toluensulfonyl chloride (363.34 mg, 1.905 mmol) was dissolved in THF, then converted to the first flask at 0ºC. The solvent was evaporated and an extraction using the diethyl ether and water. The organic part was dried over MgSO_4_, filtered, and the solvent removed under reduced pressure to give 9,10-(CH_3_-C_6_H_4_-SO_3_-CH_2_CH_2_CH_2_)_2_-1,7-*closo*-C_2_B_10_H_10_, **7**, (302 mg, 85%). ATR: ν = 3064 (s, (C_c_-H)), 2971, 2955, 2895 (s, (C_alkyl_-H)), 2596, 2565 (s, (B-H)), 1349 (s, (S=O)),1189, 1175 (s, (S-O)), 1097, 981, 954, 918 (s, (C-O)). ^13^C{^1^H} NMR (300.13 MHz, (CD_3_)_2_CO) δ: 144.77 (s, CH_3_-C_6_H_4_), 133.80 (C_6_H_4_-S), 129.94 (s, C_aryl_), 127.76 (s, C_aryl_), 72.67 (s, TsO*C*H_2_CH_2_CH_2_), 53.26 (s, *C*_c_), 37.28 (s, TsOCH_2_*C*H_2_CH_2_), 20.62 (s, C*H*_3_-C_6_H_5_), 9.78 (br s,TsOCH_2_CH_2_*C*H_2_). ^1^H-NMR (300.13 MHz, (CD_3_)_2_CO) δ: 7.82 (d, 4H, ^1^*J*(H,H) = 7.2, H_aryl_), 7.49 (d, 4H,^*1*^*J*(H,H) = 8.0, H_aryl_), 4.05 (t, ^3^*J*(H,H) = 6.6, 4H, TsO-C*H*_2_), 3.45 (br s, 2H, C_c_*-H)*, 2.47 (s, 6H, C*H*_3_C_6_H_4_), 1.75 (m, 4H, TsOCH_2_C*H*_2_CH_2_), 0.77 (m, 4H, TsOCH_2_CH_2_C*H*_2_). ^1^H{^11^B} NMR (300.13 MHz, (CD_3_)_2_CO) δ: 7.82 (d, 4H, ^1^*J*(H,H) = 7.2, H_aryl_), 7.49 (d, 4H,^*1*^*J*(H,H)= 8.0, H_aryl_), 4.05 (t, ^3^*J*(H,H)= 6.6, 4H, TsO-C*H*_2_), 3.45 (br s, 2H, C_c_*-H)*, 2.47 (s, 6H, C*H*_3_C_6_H_4_), 3.04, 2.03, 2.00 (br s, 8H, B*H*),1.75 (m, 4H, TsOCH_2_C*H*_2_CH_2_), 0.77 (m, 4H, TsOCH_2_CH_2_C*H*_2_). ^11^B NMR (96.29 MHz, (CD_3_)_2_CO) δ: 3.4 (s, 2B, B(9,10)), −5.4 (d, ^1^*J*(B,H) = 153, 2B, B(5,12)), −12.7 (d, ^1^*J*(B,H) = 155, 4B, B(4,6,8,11)), −19.2 (d, ^1^*J*(B,H) = 177, 2B, B(2,3)).

#### 3.1.6. Synthesis and Characterization of **8**

To a stirred solution of previously dried 9,10-(ClCH_2_CH_2_CH_2_)_2_-1,7-*closo*-C_2_B_10_H_10_, **5**, (95 mg, 0.319 mmol) in DMF (10 mL), NaN_3_ (314.41 mg, 4.83 mmol) was added. At room temperature, the mixture was stirred for 24 h. Then, the solvent was evaporated under vacuum and an extraction with a mixture C_6_H_5_CH_3_-H_2_O was done. After washing it several times with H_2_O, the collected organic layer was dried over MgSO_4_, filtered, and the solvent removed under reduced pressure to give 9,10-(N_3_CH_2_CH_2_CH_2_)_2_-1,7-*closo*-C_2_B_10_H_10_, **8**, (80.75 mg, 81%). ATR: ν= 3076 (C_c_-H), 2929, 2890 (C_alkyl_-H), 2591 (B-H), 2095 (C-N). ^13^C{^1^H} NMR (300.13 MHz, (CD_3_)_2_CO) δ: 53.42 (s, C-N_3_), 53.26 (C_c_-H), 11.13 (m, CH_2_). ^1^H-NMR (300.13 MHz, (CD_3_)_2_CO) δ: 3.84 (s, 2H, C_c_-H), 3.34 (t, ^3^*J*(H,H) = 6.90, 4H, N_3_*CH*_2_CH_2_CH_2_), 1.69 (m, N_3_*CH*_2_CH_2_CH_2_), 0.89 (m, 4H, N_3_*CH*_2_CH_2_CH_2_). ^1^H{^11^B} NMR (300.13 MHz, (CD_3_)_2_CO) δ: 3.84 (s, 2H, C_c_-H), 3.34 (t, ^3^*J*(H,H) = 6.90, 4H, N_3_*CH*_2_CH_2_CH_2_), 2.52, 2.23, 2.13 (br s, BH), 1.69 (m, N_3_*CH*_2_CH_2_CH_2_), 0.89 (m, 4H, N_3_*CH*_2_CH_2_CH_2_). ^11^B NMR (96.29 MHz, (CD_3_)_2_CO) δ: 1.4 (s, 2B, B(9,10)), −5.3 (d, ^1^*J*(B,H)= 157, 2B, B(5,12)), −12.6 (d, ^1^*J*(B,H) = 160, 4B, B(4,6,8,11)), −19.1 (d, ^1^*J*(B,H) = 180, 2B, B(2,3)).

#### 3.1.7. Synthesis and Characterization of **9**

To a solution of 9,10-(N_3_CH_2_CH_2_CH_2_)_2_-1,7-*closo*-C_2_B_10_H_10_, **8**, (27 mg, 0.086 mmol) in a mixture of dioxane (2 mL) and distilled H_2_O (2 mL), phenylacetylene (0.04 mL, 0.36 mmol), sodium ascorbate (17.037 mg, 0.086 mmol) and CuSO_4_.5H_2_O (21.47 mg, 0.086 mmol) were added in this order. After 20 min, a yellow solid started to be formed. The reaction was stopped after 3 h, when the walls of the small flask were full yellow solid and the solution was green. Then, the yellow solid was separated and very well dried under vacuum to give 9,10-(C_6_H_5_C_2_N_3_CH_2_CH_2_CH_2_)_2_-1,7-*closo*-C_2_B_10_H_10_, **9**, (76 mg, 86%). Elemental analysis: calc: C% 56.23, H% 6.29; exp: %C 56.69, %H 6.88. ATR: ν= 3057 (C_c_-H), 2924, 2892, 2852, 2826 (C_alkyl_-H, CH_2_N), 2588 (B-H). ^13^C{^1^H} NMR (300.13 MHz, (CD_3_)_2_SO) δ: 146.69 (C_6_H_5_-*C*-N=N-N), 131.35 (C_6_H_5_-*C*=*C*-N), 129.34 (C_6_H_5_), 128.22, 125.57 (C_6_H_5_), 121.68 (C_6_H_5_), 54.15 (C_c_-H), 52.25 (N-*CH*_2_CH_2_CH_2_), 30.68 (N-CH_2_*CH*_2_CH_2_), 11.4 (NCH_2_CH_2_*CH*_2_). ^1^H-NMR (300.13 MHz, (CD_3_)_2_SO) δ: 8.56 (s, 2H, C=*CH*-N), 7.85 (d, 4H,^1^*J* = 7.7, C_6_H_5_), 7.44 (t, 4H, ^3^*J* = 7.5, C_6_H_5_), 7.32 (t, 2H,^3^*J* = 7.1, C_6_H_5_), 3.86 (s, 2H, C_c_-H), 4.34 (m, 4H, N*CH_2_*CH*_2_*CH_2_), 1.86 (m, 4H, NCH_2_*CH*_2_CH_2_), 0.69 (m, 4H, NCH_2_CH_2_*CH*_2_). ^1^H{^11^B} NMR (300.13 MHz, (CD_3_)_2_SO) δ: 8.56 (s, 2H, C=*CH*-N), 7.85 (d, 4H,^1^*J* = 7.7, C_6_H_5_), 7.44 (t, 4H, ^3^*J* = 7.5, C_6_H_5_), 7.32 (t, 2H,^3^*J* = 7.1, C_6_H_5_), 3.86 (s, 2H, C_c_-H), 4.34 (m, 4H, N*CH_2_*CH*_2_*CH_2_), 2.38, 2.10, 2.02 (br s, B-H), 1.86 (m, 4H, NCH_2_*CH*_2_CH_2_), 0.69 (m, 4H, NCH_2_CH_2_*CH*_2_). ^11^B and ^11^B{^1^H} NMR (96.29 MHz, (CD_3_)_2_SO) δ: 1.4 (s, 2B, B(9,10)-C), −5.5 (2B, B(5,12)), −13.2 (4B, B(4,6,8,11)), −19.9 (2B, B(2,3)).

#### 3.1.8. Synthesis and Characterization of **10**

To a stirred solution of 9,10-(CH_2_=CHCH_2_)_2_-1,7-*closo*-C_2_B_10_H_10_ (150 mg, 0.67 mmol) in dry THF (10 mL) at 0 °C were added dropwise *n*-BuLi in hexane (0.98 mL, 1.5 M, 1.47 mmol), the resulting solution was stirred at 0 °C for 1 h. Then the mixture was cooled at −78°C to add dropwise a solution of CH_2_=CHCH_2_-Br in dry THF (1.54 mL, 1 M, 1.54 mmol), and allowed to stir overnight at room temperature. Afterwards, the solvent was removed and 10 mL of diethyl ether and 10 mL of HCl (0.25M) were added to the residue. The organic layer was separated from the mixture, and the aqueous layer was extracted with diethyl ether (3 × 10mL). The combined organic phase was dried over MgSO_4_, filtered, and the solvent removed under reduced pressure to give 71% of 9,10-(CH_2_=CHCH_2_)_2_-1,7-(CH_2_=CHCH_2_)_2_-*closo*-C_2_B_10_H_8_ (145 mg). ^13^C{^1^H} NMR (300.13 MHz, (CD_3_)_2_CO) δ: 140.17 (s, CH_2_=*C*H-CH_2_-B(9,10)), 133.96 (s, CH_2_=*C*HCH_2_-C(1,2)), 117.98 (s, *C*H_2_=CH-CH_2_-B(9,10)), 111.96 (s, *C*H_2_=CH-CH_2_-C(1,2)), 71.8 (s, *C*_c_), 40.97 (s, CH_2_=CH-*C*H_2_-C(1,2)), 21.48 (m, CH_2_=CH-*C*H_2_- B(9,10)). ^1^H-NMR (300.13 MHz, (CD_3_)_2_CO) δ: 5.80 (m, 2H, CH_2_=C*H-*CH_2_-B(9,10)), 5.60 (m, 2H, CH_2_=C*H*-CH_2_-C(1,2)), 4.90 (m, 4H, C*H*_2_=CH-CH_2_-C(1,2)), 4.86 (m, 4H, C*H*_2_=CH-CH_2_-B(9,10)), 2.57 (d, ^3^*J*(H-H)=7.3 Hz, 4H, CH_2_=CH-C*H*_2_-C(1,2)), 1.71 (br s, 4H, CH_2_=CH-C*H*_2_-B(9,10)). ^1^H{^11^B} NMR (300.13 MHz, (CD_3_)_2_CO) δ: 5.80 (m, 2H, CH_2_=C*H-*CH_2_-B(9,10)), 5.60(m, 2H, CH_2_=C*H-*CH_2_-C(1,7)), 4.90 (m, 4H, C*H*_2_=CHCH_2_-C(1,7)), 4.86 (m, 4H, C*H*_2_=CHCH_2_-B(9,10)), 2.57 (d, ^3^*J*(H,H) = 7.3 Hz, 4H, CH_2_=CHCH_2_-C(1,7)), 2.17 (brs, 2H, B(5,12)-*H*), 2.12 (brs, 4H, B(4,6,8,11)-*H*), 1.87 (brs, 2H, B(2,3)-*H*), 1.71 (brs, 4H, CH_2_=CH-C*H*_2_-B(9,10)). ^11^B-NMR (96.29 MHz, (CD_3_)_2_CO) δ: 0.8 (s, 2B, B(9,10)), −5.4 (d, ^1^*J*(B,H) = 154, 2B, B(5,12)), −10.2 (d, ^1^*J*(B,H) = 159, 4B, B(4,6,8,11)), −15.9 (d, ^1^*J*(B,H) =174, 2B, B(2,3)).

#### 3.1.9. Synthesis and Characterization of **11**

To a stirred solution of 9,10-(CH_2_=CHCH_2_)_2_-1,7-(CH_2_=CHCH_2_)_2_-*closo*-C_2_B_10_H_8_, **3**, (130 mg, 0.24 mmol) in THF (3.5 mL) at 0 °C, was added, drop wise, a solution of BH_3_·THF in THF (1.765 mL, 1M, 1.765 mmol). The resulting suspension was stirred at 0 °C for 30 min and at room temperature for further 30 min. Then, the reaction mixture was cooled again to 0 °C in an ice-water bath and water (1mL) was slowly added. When gas evolution had stopped, an aqueous KOH solution (0.583 mL, 3 M, 1.75 mmol) and subsequently, H_2_O_2_ in water (0.3 mL, 35%), were added. Stirring was maintained at room temperature for 1.5 h, after which two liquid phases were observed. The upper organic layer was separated from the mixture and the aqueous layer. Then, it was washed with THF (3 × 2 mL). The combined organic phase was dried over MgSO_4_, filtered and the solvent removed in vacuo to give 9,10-(HOCH_2_CH_2_CH_2_)_2_-1,7-(HOCH_2_CH_2_CH_2_)_2_-*closo*-C_2_B_10_H_8_.Yield: 119mg (74%). Elemental analysis: calc: C% 44.7, H% 9.6; exp: %C 45.1, %H 9.8. ^13^C{^1^H} NMR (300.13 MHz, (CD_3_)_2_CO) δ: 66.04 (s, HOCH_2_CH_2_CH_2_-*C*(1,7)), 64.18 (s, HO*C*H_2_CH_2_CH_2_-C(1,7), HO*C*H_2_CH_2_CH_2_-B(9,10)), 60.52 (s, HOCH_2_CH_2_CH_2_-*C*(1,7)), 33.38 (s, HOCH_2_*C*H_2_CH_2_-C(1,7), 33.19 (s, HOCH_2_*C*H_2_CH_2_-B(9,10)), 23.83 (s, HOCH_2_CH_2_*C*H_2_B(9,10)). ^1^H{^11^B} NMR (300.13 MHz, (CD_3_)_2_CO) δ: 3.67–3.51 (m, 12H, *H*OC*H*_2_CH_2_CH_2_), 1.59 (m, 8H, HOCH_2_C*H*_2_CH_2_), 0.99–0.81 (m, 12H, HOCH_2_CH_2_C*H*_2_). ^11^B-NMR (96.29 MHz, (CD_3_)_2_CO) δ: 1.2 (s, 2B, B(9,10)), −5.6 (d,^1^*J*(B,H) = 147, 2B, B(5,12)), −10.6 (d, ^1^*J*(B,H) = 150, 4B, B(4,6,8,11)), −15.8 (d, ^1^*J*(B,H) = 175, 2B, B(2,3)).

#### 3.1.10. Synthesis and Characterization of **12**

To a stirred solution of 9,10-(CH_2_=CHCH_2_)_2_-1,7-*closo*-C_2_B_10_H_10_, **3**, (20 mg, 0.09 mmol) in THF (3 mL) at 0 °C, was added, drop wise, a solution of BuLi (0.18 mmol, 1.6 M, 0.12 mL). The resulting suspension was stirred at 0 °C for 30 min and at room temperature for further 30 min. In another flask and under nitrogen, 8-{3,3’-Co(8-C_4_H_8_O_2_-1,2-C_2_B_9_H_10_)(1’,2’-C_2_B_9_H_11_) (0.18 mmol, 77 mg) was dissolved in 8 mL of THF. Then, the new solution was transferred to the suspension mixture and the reaction mixture stirred 2 h under reflux at inert atmosphere. The solvent was evaporated, and an extraction took place. The organic layer was evaporated to give a mixture of white and orange compounds. The 8-{3,3’-Co(8-C_4_H_8_O_2_-1,2-C_2_B_9_H_10_)(1’,2’-C_2_B_9_H_11_) was recuperated and the new isomer 9,10-(CH_3_CH=CH)_2_-1,7-*closo*-C_2_B_10_H_10_, 12, was obtained with 80% of yield (16 mg). ATR: ν = 3046 (C_c_-H), 2998–2849 (m, υs(CH_3_, =CH)), 2623, 2592 (vs, υs(B-H)), 1634, 1442 (vs, υ_s_(C=C)), 978 (s, υ_as_(-CH=CH-)). ^1^H-NMR (300.13 MHz, CDCl_3_) δ: 5.90 (m, 2H, CH_3_C*H*=CH), 5.55 (m, 2H, CH_3_CH=C*H*), 2.82 (s, 2H, C_c_-*H*), 1.78 (dd, ^3^*J*(H_a_,H_b_) = 6.3, *^4^J*(H_a_,H_c_) = 1.6, 6H, C*H*_3_CH=CH). ^1^H{^11^B} NMR (300.13 MHz, CDCl_3_) δ: 5.90 (m, 2H, CH_3_C*H*=CH), 5.57 (m, 2H, CH_3_CH=C*H*), 2.82 (s, 2H, C_c_-*H*), 2.50 (s, 2H, B(5,12)-*H*), 2.34 (s, 2H, B(2,3)-*H*), 2.19 (s, 4H, B(4,6,8,11)-*H*), 1.78 (d, 6H, C*H*_3_CH=CH). ^11^B-NMR (96.29 MHz, (CD_3_)_2_CO) δ: −0.5 (s, 2B, B(9,10)), −5.8 (d,^1^*J*(B,H) = 159, 2B, B(5,12)), −12.5 (d, ^1^*J*(B,H) = 163, 4B, B(4,6,8,11)), −19.8 (d, ^1^*J*(B,H) = 180, B(2,3)). Good crystals suitable for X-ray diffraction were grown in acetone. 

### 3.2. X-ray Structure Determinations of 9,10-(HOCH_2_CH_2_CH_2_)_2_-1,7-closo-C_2_B_10_H_10_, ***4*** and 9,10-(CH_3_CH=CH)_2_-1,7-closo-C_2_B_10_H_10_, ***12***


Single-crystal data collections for **4** and **12** were performed with an Bruker D8 QUEST ECO three-circle diffractometer system equipped with a Ceramic x-ray tube (Mo Kα, λ = 0.71076 Å) and a doubly curved silicon crystal Bruker Triumph monochromator (Bruker, Karlsruhe, Germany). The structures were solved by direct methods and refined on F^2^ by the SHELXL97 program [65]. The non-hydrogen atoms were refined with anisotropic displacement parameters. The hydrogen atoms were treated as riding atoms using the SHELXL97 default parameters. The crystallographic, structure refinement, and bond parameters for **4** and **10** are reported in CIF-files deposited at CCDC with the reference numbers CCDC 2004945 and 2004946. These data can be obtained free of charge via www.ccdc.cam.ac.uk/conts/retrieving.html (or from the Cambridge Crystallographic Data Centre, 12 Union Road, Cambridge CB2 1EZ, U.K.; fax: +44-1223-336033; or e-mail: deposit@ccdc.cam.ac.uk).

### 3.3. Hirshfield Surface Analysis

The Hirshfeld surface analyses were run using the CIF format by the CrystalExplorer program [62]. Hirshfeld surface analysis help to recognize the strong and weak intermolecular interactions area and the nature of these interactions from the electron distribution. The d_norm_ (normalized contact distance) is given by the Equation (1):(1)$$d_{norm}=\frac{d_{i}+r_{i}^{vdw}}{r_{i}^{vdw}}+\frac{\;d_{e}+r_{e}^{vdw}}{r_{e}^{vdw}}$$
where d_i_ is from the Hirshfeld surface to the nearest atom outside-external, d_e_ from the Hirshfeld surface to the nearest internal atom, and r^vdw^ is the Van Der Walls radii of the atom

## 4. Conclusions

All allyl di and tetrabranched derivatives of the *m*-carborane framework have been synthesized. The starting 9,10-(allyl)_2_-1,7-*closo*-carborane compound was made by Kumada cross-coupling reaction on 9,10-I_2_-1,7-*closo*-carborane with allyl Grignard reagent in the presence of Pd(II) and Cu(I) as catalysts. These olefin groups have led to a variety of functional groups, alcohol, chloro, tosyl, and azide that have permitted to produce esters and 1,2,3-triazoles by the azide-alkyne cycloaddition, as examples of reactions that show the wide possibilities of this globular icosahedral *m*-carborane to act as a novel core for periphery-decorated macromolecules. Importantly, the four branches in the tetrabranched *m*-carborane derivatives are located in two perpendicular planes and are coplanar in the *o*-carborane isomer. This difference provides novel cores for 3D and 2D radially grown periphery-decorated macromolecules, respectively. Unexpectedly, the isomerization of B-allyl to B-propenyl vertexes in 9,10-(allyl)_2_-1,7-*closo*-C_2_B_10_H_10_ was observed in THF. DFT calculation studies conclude that the comparable acidity of the allyl groups and the C_c_-H of the *m*-carborane unit allows a deprotonation/protonation isomerization of the allyl group as it is well known for allylbenzenes. X-ray crystal structures of 9,10-(OHCH_2_CH_2_CH_2_)_2_-1,7-*closo*-C_2_B_10_H_10_ and 9,10-(CH_3_CHCH)_2_-1,7-*closo*-C_2_B_10_H_10_ compounds show an extensive network of hydrogen bonding and π···H-C_c_ contacts, respectively, due to the presence of alcohol and olefin groups that have been analyzed by Hirshfeld surfaces and decomposed fingerprint plots.

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
