# Peer review of "m-Carborane as a Novel Core for Periphery-Decorated Macromolecules"

_molecules, 2020, doi:10.3390/molecules25122814_

Round 1

Reviewer 1 Report

This manuscript by Vinas et al. presents allyl di and tetrabranched derivatives of the meta-carborane framework. The starting 9,10-(allyl) 2 -1,7-closo-carborane compound was made by Kumada cross coupling reaction from 9,10-diiodinated meta-carborane with allyl Grignard reagent in the presence of Pd(II) and Cu(I) catalysts. These olefin groups have led to a variety of functional groups including alcohol, chloro, tosyl, and azide that have permitted to produce esters and 1,2,3-triazoles by the azide-alkyne cycloaddition. Interestingly, isomerization of B-allyl to B-propenyl vertexes in 9,10-(allyl)2-1,7-closo-C2B10H10 was observed in THF and DFT calculations support comparable acidity of the allyl groups and the cage C-H moieties of the meta-carborane unit to explain this phenomenon. X-ray ray crystal structures of the bisallyl and hydroxylated boron cages show an extensive network of hydrogen bonding and π···H-C contacts.

All compounds were throughly characterised including 11B-11B 2D-NMR spectroscopy. The paper can be published without any further experiments.

However, I am a bit concerned about the quality of presentation. Some phrases sound awkward and the consistency throughout the article must be improved.

Examples:

i) The terms o- and m- for ortho and meta should always be italic, mostly they are but not so in lines 90, 143, 210, 211, 217, ... please correct.

ii) When acidities (e.g.  C-H acidity) is discussed, please, state and cite the correct pka value form the literature. This makes the discussion more specific.

iii) Some phrases sound non-scientific to me: Examples:

   Line 40: First and foremost, ...

   Line 46: ..., the functionalisation does not take place this way.

   Line 50: On top of that, ...

iv) In the experimental part the compound numbers should all be bold. I also find the term "To a stirring solution ... " awkward and would replace it by the the term "stirred solution"

v) Compound 4: line 461 - the yield should not be given as 86.74%.

vi) Elemental analysis results are inconstently given with one or two decimals digits, sometimes for the same compound. Please correct consitently.

vii) In the electronic supporting information the solvent must be stated for each NMR-spectrum. It was tedious for me to look it up for each compound.

viii) Reference 62. The name of "Mark A. Spackman" to cite the Crystal Explorer is misspelled. Please correct.

Author Response

Referee 1

Comments and Suggestions for Authors

However, I am a bit concerned about the quality of presentation. Some phrases sound awkward and the consistency throughout the article must be improved. Examples:

  1. i) The terms o- and m- for ortho and meta should always be italic, mostly they are but not so in lines 90, 143, 210, 211, 217, ... please correct.

Referee is right that the isomers’ term should be in italics. We have amended them through the manuscript in its revised file.

  1. ii) When acidities (e.g.  C-H acidity) is discussed, please, state and cite the correct pka value form the literature. This makes the discussion more specific.

We have added the required information along with the corresponding references. Please, see the added text on page 5, lines 175-179.

iii) Some phrases sound non-scientific to me: Examples:

   Line 40: First and foremost, ...

   Line 46: ..., the functionalisation does not take place this way.

   Line 50: On top of that, ...

  1. iv) In the experimental part the compound numbers should all be bold. I also find the term "To a stirring solution ... " awkward and would replace it by the the term "stirred solution"

All these remarks have been amended.

  1. v) Compound 4: line 461 - the yield should not be given as 86.74%.
  2. vi) Elemental analysis results are inconstently given with one or two decimals digits, sometimes for the same compound. Please correct consitently.

The decimal digits on yields and elemental analysis results have been corrected in the revised manuscript file.

vii) In the electronic supporting information the solvent must be stated for each NMR-spectrum. It was tedious for me to look it up for each compound.

The deuterated solvent has been added in the S.I.

viii) Reference 62. The name of "Mark A. Spackman" to cite the Crystal Explorer is misspelled. Please correct.

The mistake on the author family name has been corrected.

Reviewer 2 Report

This paper can be of help to researchers who want to study on the reaction of introducing substituents to the boron and/or carbon atoms of meta-carboranes. Molecular structure analysis was performed using X-ray crystallography, and Hirshfield Surface Analysis was used to help understand the interaction between molecules. However, there are too many typos in the text, and I suggest that the authors carefully review them. The publication in Molecules is thus recommended with minor revision.

Author Response

Referee 2

Comments and Suggestions for Authors

This paper can be of help to researchers who want to study on the reaction of introducing substituents to the boron and/or carbon atoms of meta-carboranes. Molecular structure analysis was performed using X-ray crystallography, and Hirshfield Surface Analysis was used to help understand the interaction between molecules. However, there are too many typos in the text, and I suggest that the authors carefully review them. The publication in Molecules is thus recommended with minor revision.

Thanks to the referee for his/her manuscript evaluation. We have reviewed the text and corrected the typos found.

Reviewer 3 Report

In this submission, the authors focus attention on the less well-studied meta-carborane system. Their synthetic achievements include the incorporation of esters, sulfonates, azides and triazoles, and these novel compounds have been fully characterized spectroscopically and, in some cases, by X-ray crystallography. I was particularly pleased by the use of 2D 11B-11B NMR techniques, but I note that throughout the manuscript this is designated as RMN, and should be corrected.

The experimental work is complemented by computational investigations, not only on the new products, but also making comparisons with the more widely-studied ortho-carboranes.

Overall, this is a valuable addition to the carborane field, and will undoubtedly please Todd Marder in whose honour it is being submitted.

Author Response

Referee 3

Comments and Suggestions for Authors

In this submission, the authors focus attention on the less well-studied meta-carborane system. Their synthetic achievements include the incorporation of esters, sulfonates, azides and triazoles, and these novel compounds have been fully characterized spectroscopically and, in some cases, by X-ray crystallography. I was particularly pleased by the use of 2D 11B-11B NMR techniques, but I note that throughout the manuscript this is designated as RMN, and should be corrected.

Thanks to the referee for his/her comments. We have searched for the word RMN and found at the S.I. The word RMN has been changed by NMR over the file and the new S.I. file submitted.